# Explainable, Steerable Models with Natural Language Parameters and Constraints

## Abstract

Statistical modeling can uncover patterns in large datasets, but these patterns may not be explainable or relevant to our specific interest. For example, Gaussian mixture models are commonly used to explore text corpora, but it is hard to explain what each cluster means or steer them to use specific attributes (e.g. cluster based on style but not topic). To improve explainability and steerability, we introduce models where parameters are represented as *natural language strings*. For example, instead of using a Gaussian to represent a cluster, we represent it with a natural language predicate such as "*has a casual style*". By leveraging the denotational semantics of natural language, we interpret these predicates as binary feature extractors and use them as building blocks for classical statistical models such as clustering, topic modeling, and regression. Language semantics also lets us specify constrains on the learned string parameters, such as "*the parameters should be style-related*". To learn in our framework, we propose an algorithm to optimize the log-likelihood of these models, which iteratively optimizes continuous relaxations of string parameters and then discretizes them by explaining the continuous parameters with a language model. Evaluating our algorithm across three real corpora and four statistical models, we find that both the continuous relaxation and iterative refinement to be crucial. Finally, we show proof-of-concept applications in controllably generating explainable image clusters and describing major topic variations across news from different months.

## 1 Introduction

To discover and explain patterns in a dataset with structured modalities such as texts or images, researchers often first fit a statistical model on the dataset and then interpret the learned models. For example, n-gram logistic regression (Wang & Manning, 2012) is frequently used to explain differences between text distributions; and unsupervised models such as image clustering (Caron et al., 2018) and topic modelling (Blei et al., 2003) are used to explore datasets in machine learning (Sivic et al., 2005), social sciences (Nguyen et al., 2020), and bioinformatics (Liu et al., 2016).

However, these models are usually hard to explain and steer. Consider text clustering as an example, where a data scientist has access to a corpus of social media posts and wishes to cluster them based on style. A neural text clustering algorithm (Aharoni & Goldberg, 2020) might cluster them based on topic information, since it cannot be steered by the specified constraint; it also outputs each cluster as a Gaussian distribution over neural embeddings, which is not immediately explainable to humans. More broadly, many statistical models in structured modalities lack explainability and steerability, since they rely on high-dimensional parameters (e.g. neural embeddings or word vectors), which is an inconvenient interface for humans to steer the models and for the models to explain to humans.

To improve explainability and steerability, we propose natural language as an alternative interface between human practitioners and statistical models. We introduce a framework for defining models "with **n**atural **l**anguage constraints and parameters" (NLCP), where the models are explainable because they learn natural language parameters and are steerable because we can constrain the parameters via natural language. Our core insight is that we can use the denotational semantics of a natural language predicate to extract a 0/1 feature by checking whether it is true on a sample. For instance, given the predicate $\phi = $ "*has a casual style*", its denotation $[\![\phi]\!]$ is a binary function that

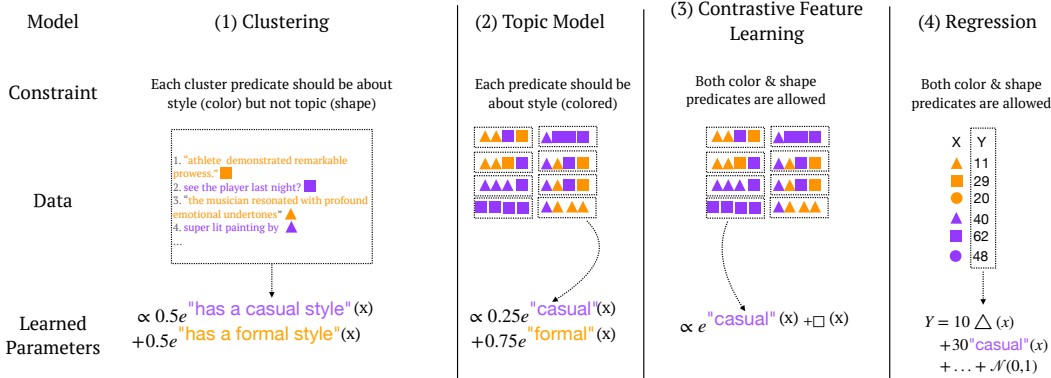

Figure 1: We illustrate four models under NLCP. (1) clustering: corpus is modelled as a mixture of distributions, each parameterized with a natural language string "*has a . . . style*" satisfying the given constraint. (2) topic model and (3) contrastive feature learning: the user provides a set of corpus, and we model each corpus either as a mixture of text distributions or a single distribution parameterized by multiple predicates. (4) regression: each provided sample $(x)$ is associated with a real-valued target $(y)$ and our goal is to explain the target value with a few predicates and coefficients.

evaluates to 1 on texts $x$ written in a causal style and 0 otherwise. For example,

$$[\![\text{"has a casual style"}]\!](\text{"with more practice we've got a shot at the title."}) = 1. \quad (1)$$

Similarly, denotational semantics allows us to check whether the parameters satisfy a constraint:

$$[\![\text{"the parameter is style-related"}]\!](\text{"has a topic of sports"}) = 0. \quad (2)$$

By using natural language to extract 0/1 features from samples and checking the constraint on the parameters, we extend a variety of classical statistical models, including clustering, topic-models, and regression (Figure 1).

Fitting a model requires optimizing the log-likelihood, but this is challenging under NLCP because natural language parameters are discrete and thus do not admit gradient-based optimization. To address this, we present a general framework for optimizing NLCP models. Our framework uses gradient descent to optimize a continuous relaxation of natural language parameters, where we simulate denotation as a dot product between a continuous predicate and some embedding of $x$. It then discretizes the continuous predicate by prompting a language model to generate candidate language predicates that explain the variation in the learned continuous predicate. We alternate between continuous optimization and discretization to iteratively refine the learned natural language parameters.

To evaluate our optimization algorithm, we create statistical modelling problems where the optimal natural language parameters are known and can be used as references for the learned parameters. We evaluated on four different statistical models (clustering, topic modelling, contrastive feature learning, and regression) and used three different real datasets (NYT articles, AG-News, and DB-Pedia (Sandhaus, 2008; Zhang et al., 2015)), where each text sample is annotated with a gold topic or location description. We found that both continuous relaxation and iterative refinement improve performance. Finally, we show proof-of-concept application in controllably generating explainable image clusters and describing major topic variations across months in detail, showing that our framework could be potentially extended to other modalities and unlock new applications.

## 2   RELATED WORK

**Statistical Modelling in Text.** Statistical models based on n-gram features or neural embeddings are broadly used to analyze text datasets. For example, logistic regression or naive bayes models are frequently used to explain differences between text distributions (Wang & Manning, 2012); Gaussian mixture on pre-trained embeddings can create text clusters (Aharoni & Goldberg, 2020); topic models can mine major topics across a large collection of documents (Blei et al., 2003) across time (Blei & Lafferty, 2006). However, since these models usually rely on high-dimensional parameters, they are difficult to explain (Chang et al., 2009) or steer (Aharoni & Goldberg, 2020).

To explain these models, Carmel et al. (2009); Treeratpituk & Callan (2006); Zhang et al. (2018) propose to explain each topic or clusters by extracting candidate phrases either from the corpus or from Wikipedia. To steer these models, Hu et al. (2014) allow users to shape the clusters by specifying words that should co-occur in a topic. Our work proposes a complementary approach to explain and steer models with natural language, which is potentially more flexible and convenient.

**Modelling with Latent Interpretable Structures.** Many prior works have tried to train models to accomplish complex tasks with interpretable latent structures. For example, Neural Module Networks aim to decompose a problem into interpretable subparts and then solve them with individual modules (Andreas et al., 2016; Subramanian et al., 2020); Concept Bottleneck Models (Koh et al., 2020) aim to learn interpretable intermediate features. Conceptually closest to our work, Andreas et al. (2018) learns discrete language parameters to improve generalization. Our modelling framework focuses on explaining datasets rather than improving predictive performance, and we enforce explainability by learning natural language parameters and a small set of real-valued real parameters.

**LLM for Exploratory Analysis.** Due to its code generation capability (Chen et al., 2021), large language models have been used to automatically generate programs to analyze a dataset and generate reports from them (Ma et al., 2023; Hassan et al., 2023). In comparison, our work focuses on generating natural language parameters to extract real-valued features from structured data.

**Inductive Reasoning with Language Models.** Prior works have prompted the language model with raw data and ask it to perform inductive reasoning. For example, Zhong et al. (2022; 2023) explain distributional differences, Singh et al. (2022) explains relation between tuples, Wang et al. (2023) generates explainable clusters, Yang et al. (2022) extracts new knowledge by reasoning about hidden assumptions, Tong et al. (2023) categorizes failure modes of a multi-modal model, Yang et al. (2023) proposes hypotheses for open-ended scientific questions, and Singh et al. (2023); Bills et al. (2023) explain neural network representations. Our work formalizes a framework to define more complex models with natural language explanations by combining it with classical statistical models.

**Discrete Prompt Optimization.** Many prior works have tried to optimize discrete prompts to improve the predictive performance of a model (Shin et al., 2020; Deng et al., 2022), and some recent works demonstrated that language models can optimize prompts to reach state-of-the-art accuracy (Zhou et al., 2022; Yang et al., 2023). In comparison, our work focuses on optimizing prompts to explaining patterns unknown to the user.

## 3 MATHEMATICAL FORMULATION

We will define statistical models $p$ as generative distributions over some space $\mathcal{X}$ (e.g. all images or all text strings of a certain length). Our framework, NLCP, defines generative distributions in 3 sets of parameters: natural language predicates $\phi$, weights $w$ on those predicates, and a base distribution $B(x)$. We will first show how to define a basic exponential family model using NLCP (Section 3.1), then use this as a building block for more complex objectives such as clustering and topic modelling.

### 3.1 NOTATIONS

The fundamental building block of NLCP is a generative distribution $p(x|w, \vec{\phi})$ over $\mathcal{X}$. The distribution $p(x|w, \vec{\phi})$ is parameterized by three variables: $\vec{\phi}$, a list of learned natural language predicates; $w$, a learned real vector, which reweights and sums the feature values; and $B$, an implicit base distribution, which reweights the probability of a text sample.

The core idea in NLCP is a way of interpreting a natural language predicate $\phi$ as a logical predicate $[\![\phi]\!] : \mathcal{X} \to \{0, 1\}$, via a "denotation operator" $[\![\cdot]\!]$. Given $[\![\cdot]\!]$, the generative distribution $p$ is an exponential family with base distribution $B$ and feature function $[\![\vec{\phi}]\!]$:

$$p(x|w, \vec{\phi}) \propto B(x)e^{w^T [\![\vec{\phi}]\!](x)}. \tag{3}$$

We next explain $\vec{\phi}$, $[\![\cdot]\!]$, $p(x|w, \vec{\phi})$, and $B$ in greater detail.

**Explainable Feature Functions via Natural Language Parameters.** A natural language predicate $\phi$ is a string and its denotation $[\![\phi]\!]$ maps samples $x \in \mathcal{X}$ to $\{0, 1\}$. For example, if $\phi = $ "*has a casual style*", then $[\![\phi]\!]($ "*we've got a shot.*"$) = 1$. Since a models typically require multiple features

to explain the data, we consider vectors $\vec{\phi}$ of $K$ predicates, where now $[\![\vec{\phi}]\!]$ maps $\mathcal{X}$ to $\{0,1\}^K$:

$$[\![\vec{\phi}]\!](x) := \left([\![\phi_1]\!](x), [\![\phi_2]\!](x), \ldots, [\![\phi_K]\!](x)\right) \tag{4}$$

To instantiate $[\![\cdot]\!]$ computationally, we prompt a language model to ask whether $\phi$ is true of the input $x$, following practice from prior works (Zhong et al., 2022; 2023). See prompt in Figure 3.

**Modelling Text Distributions with Features.** We provide a few simple examples of the generative distribution $p$. First, we can create a uniform distribution over all sports-related texts in $\mathcal{X}$ by up-weighting $x$ based on $\phi = $ *"has a topic of sports"*. This can be used to represent a topic or a cluster (see Section 3.2 for examples). Formally,

$$p(x|[\tau], [\text{"has a topic of sports."}]) \propto e^{\tau [\![\text{"has a topic of sports."}]\!](x)}, \tag{5}$$

where $w$ consists of a single temperature parameter $\tau$ that is usually taken to $\infty$.

Beyond forming clusters, we can combine multiple predicates to model more complex distributions; e.g. the following distribution places higher weight on non-formal sports-related texts:

$$p(x|w = [-5, 3], \vec{\phi} = [\text{"has a formal style"}, \text{"has a topic of sports"}]) \tag{6}$$

**Modelling Uninformative Nuisances with a Base Distribution.** Suppose we are given a set of news articles and want to model the distribution over its sports-related subset $p_{\text{sports}}$. We would ideally like to model this with a single predicate *"has a topic of sports"* as in equation 5. However, this distribution would contain all sports-related text (not just news articles), e.g. sports-related tweets and conversations. In general, modelling $p_{\text{sports}}$ requires additional predicates that capture its difference with a typical sample from $\mathcal{X}$, which include many uninformative properties such as whether $x$ is grammatical or in English, distracting us from the true goal to explain that $p_{\text{sports}}$ is sports-related.

Ideally, we would model these uniformative differences without using natural language predicates, since we want the predicates to explain informative properties rather than background information. To do this, we introduce a base distribution $B$ over $\mathcal{X}$ to reweight each sample, so that $\vec{\phi}$ can focus on differences from the base distribution. $B$ is closely related to the concept of "presupposition" in linguistics, which incorporates the implicit assumptions of our model.

The background distribution $B$ is thus important, but only for the purposes of selecting good $w$ and $\vec{\phi}$. We therefore take a semiparametric approach: assuming we have samples from $B$ (in the example above, some background corpus of news articles), we approximate it by its empirical distribution $\hat{B}$.

**Steering $\vec{\phi}$ with Natural Language Constraints.** We often want to steer the predicates $\vec{\phi}$ to focus on aspects that are important for a given application. To do so, we include constraints $c$ that should hold for a predicate in $\phi$. As with $\phi$, each constraint $c$ is a natural language predicate and its denotation $[\![c]\!]$ maps predicates $\phi$ to $\{0,1\}$. For example, if $c = $*"the predicate should be style-related"*, then $[\![c]\!]($*"has a topic of sports."*$) = 0$. Given a $K$-dimensional vector $\vec{c}$ of natural language constraints, define $[\![\vec{c}]\!](\vec{\phi}) = 1$ if all constraints are satisfied, and 0 otherwise, i.e.

$$[\![\vec{c}]\!](\vec{\phi}) := \forall k, [\![c_k]\!](\phi_k) = 1. \tag{7}$$

We steer the learned variable $\phi_k$ to satisfy $c_k$ by following the practice of Zhong et al. (2023): including $c_k$ in the prompt when asking an LM to generate candidate predicates for $\phi_k$.

## 3.2 Example Statistical Models under NLCP

We define four statistical models under the NLCP framework: contrastive feature learning, regression, topic modelling, and clustering. For each model, we explain its input, the learned parameters $\vec{\phi}$ and $w$, the formal log-likelihood loss function $\mathcal{L}$, and its relation to classical models.

**Contrastive Feature Learning (CFL).** The input to this task is a set of $N$ corpus $X_{1\ldots N}$. CFL learns a small set of shared feature predicates $\vec{\phi}$ along with corpus-specific weights $w_n$, modeling each corpus as an exponential family as in Section 3.1. Formally, we define the loss

$$\mathcal{L}(\vec{\phi}, w) := -\sum_{n=1}^N \sum_{x \in X_n} \log p_n(x); \quad p_n(x) := p(x|w_n, \vec{\phi}) \propto \hat{B}(x) \exp(w_n^\top \vec{\phi}(x)) \tag{8}$$

We take $\hat{B}$ to be uniform over $X_{\text{all}} = X_1 \cup \cdots \cup X_N$. Therefore, the predicates $\vec{\phi}$ should describe how each $X_n$ is different from $X_{\text{all}}$, which is why we call it contrastive feature learning.

**Topic Model.** In standard topic models (Blei et al., 2003), the input is $N$ documents, where each document is represented as a bag of words; it learns a set of topics, where each topic is represented as a distribution over words, and each document is modelled as a probabilistic mixture over topics. We can define an analogous model under NLCP: the input is $N$ corpora, where each corpus is a represented as a bag of samples $x \in \mathcal{X}$; we learn a set of $K$ topics, where each topic is a distribution over samples that satisfy a natural language predicate $\phi_k$, and each corpus is modelled as a probabilistic mixture $w_n$ over $K$ topics. Formally, we define $\mathcal{L}$ by modelling $p_n$ as follows:

$$\mathcal{L}(\vec{\phi}, w) := -\sum_{n=1}^{N} \sum_{x \in X_n} \log p_n(x); \quad p_n(x) := \sum_{k=1}^{K} w_n(k) p(x|[\tau], [\phi_k]); \tau \to \infty \quad (9)$$

As in CFL, we set $B$ to be uniform over the union of all corpora.

**Clustering.** The input is a corpus $X$ and we set $B$ to be uniform over $X$. Our model produces a set of $K$ clusters, each parameterized by a natural language predicate. As in K-means clustering, we maximize the total log-probability after assigning each sample to its most likely cluster:

$$\mathcal{L}(\vec{p}) = -\sum_{x \in X} \log(\max_k p_k(x)); \quad p_k(x) := p(x|[\tau], [\vec{\phi}_k]), \tau \to \infty \quad (10)$$

Note that we did not marginalize the probability across all clusters as in a Gaussian mixture model: a degenerated optimal solution would be $K$ identical predicates that are always true on all samples. Appendix D explains why such an issue occurs with natural language parameters in more detail.

**Regression.** The input to a regression model is a set of pairs of $(x_{n=1\ldots N}, y_{n=1\ldots N})$, where $x \in \mathcal{X}$ is the independent variable and $y \in \mathbb{R}^d$ is the dependent variable. We learn $\vec{\phi}$ and a weight matrix $w \in \mathbb{R}^{d \times K}$ such that $w^T [\![\vec{\phi}]\!](x)$ can predict the corresponding target $y$. Formally,

$$\mathcal{L}(\vec{\phi}, w) = \sum_{n=1}^{N} ||w[\![\vec{\phi}]\!](x) - y||_2^2, \quad \text{since } y \sim \mathcal{N}(w[\![\vec{\phi}]\!](x), I); \quad (11)$$

We do not need to set $B$ in this model since it predicts real vectors rather than texts.

## 4 OPTIMIZATION ALGORITHM

For each model above, we have defined the loss function $\mathcal{L}$ with respect to the parameters $\vec{\phi} \in \Phi^K$ and $w$. To fit the model, we minimize the loss under the constraint $\vec{c}$. Formally,

$$w_*, \vec{\phi}_* = \text{argmin}_{w, \vec{c}(\vec{\phi})=1} \mathcal{L}(\vec{\phi}, w) \quad (12)$$

However, such an optimization problem is challenging: since $\vec{\phi}$ is a discrete vector that needs to satisfy $\vec{c}$, it cannot be directly optimized by gradient-based methods. This section will present an algorithmic framework that optimizes the loss function of a general model under NLCP with minimal model-specific algorithmic design. Our algorithm relies on two core functions.

1. `OptimizeWandPhi`: a user-provided function that optimizes $w$ and a continuous relaxation $\tilde{\phi}_k$ for each predicate variable $\phi_k$. This can typically be implemented by simple SGD.
2. `Discretize`: a function that is given the continuous predicate $\tilde{\phi}_k$ and produces proposals for discrete predicates $\phi_k$. We provide a particular implementation below.

To make notations simpler, we omit the $\vec{\cdot}$ superscript.

**OptimizeWandPhi.** To make optimization more tractable, we first replace each discrete predicate $\phi_k$ with a continuous unit vector $\tilde{\phi}_k \in \mathbb{R}^d$. We define $[\![\tilde{\phi}_k]\!]$ as the function that maps $x$ to $\tilde{\phi}_k^\top e_x$, where $e_x$ is a feature embedding of the sample $x$ (e.g. the last-layer activations of some neural network). Thus for instance, in the exponential family model from Section 3.1, $\exp(w^\top [\![\vec{\phi}]\!](x))$ is replaced by $\exp(w^\top \tilde{\Phi} e_x)$, where $\tilde{\Phi}$ is a matrix with rows $\tilde{\phi}_k^\top$. As a result, $\mathcal{L}$ becomes differentiable with respect to $\tilde{\phi}_k$ and can typically be optimized with gradient descent.[1]

---

[1] In the clustering and topic model we set $\tau$ to be a large value (e.g. 5) to make the loss still differentiable.

Given this relaxation, we require `OptimizeWandPhi` to optimize $w$ together with the $\tilde{\phi}_k$. In fact, we require something slightly stronger: `OptimizeWAndPhi` can optimize any subset of the $w$ and $\tilde{\phi}_k$ conditional on the values of the other variables. For example:

$$\hat{\mathcal{L}}, \hat{w}, \tilde{\phi} = \texttt{OptimizeWandPhi}(\text{Empty\_Set}); \quad \hat{\mathcal{L}}, \hat{w}, \tilde{\phi}_k = \texttt{OptimizeWandPhi}(\phi_{-k}) \quad (13)$$

`OptimizeWandPhi` is different for each model and hence needs to be provided by the user, but this is straightforward in most cases. For example, under the regression loss in Equation 11, $\mathcal{L}$ is a quadratic function of $\tilde{\phi}_k$ and $w$ and can thus be optimized by gradient descent.

$$\mathcal{L}(\tilde{\phi}_k, w) = \sum_{n=1}^{N} ||w_{\cdot, -k}\phi_{-k}(x) + w_{\cdot, k}\tilde{\phi}_k \cdot e_x||_2^2. \quad (14)$$

Finally, we discuss three different choices of the embedding $e_x$. One naive approach is to use one-hot embeddings, where the embedding of each sample is orthogonal to all the others. One can also embed $x$ with a pretrained model and then normalize it to create a unit vector. We can further improve $e_x$ with "projection": only maintain and embed the constraint-related information. For example, if $x$ = "*we've got a shot*" and the constraint $c$ = "*cluster based on style*", then we first use a language model to extract the constraint-related information $\Omega_c(x)$ = "*casual; conversational*" and then embed $\Omega_c(x)$ with a pre-trained model. Appendix Figure 3 shows our prompt for projection.

**Discretize.** We now convert the continuous representation $\tilde{\phi}_k$ into a discrete predicate $\phi_k$. Our core intuition is to find a predicate $h \in \Phi$ such that its denotation is the most similar to the relaxed denotation $\tilde{\phi}_k \cdot e_x$. Concretely, we draw samples $x \sim B(x)$ and sort them based on their dot product $\tilde{\phi}_k \cdot e_x$. We then prompt a language model to generate explanations of what types of samples achieve higher dot product values than others, with the constraint that the explanations satisfy $c_k$ (Figure 3 "discretizer prompt"). We use these explanations as candidates for the predicate $\phi_k$.

Language models can generate a large number of such explanations when prompted. For computational reasons we want to cheaply filter to a subset $H$ of $M$ candidates that are likely to have low loss $\mathcal{L}$. We generate $H$ by selecting the top-$M$ predicates $h$ with the highest pearson-r correlations between $h(x)$ and $\tilde{\phi}_k \cdot e_x$ on $B$. Formally,

$$H = \texttt{Discretize}(\tilde{\phi}_k, c_k, M). \quad (15)$$

Given `OptimizeWandPhi` and `Discretize`, our overall algorithm first initializes the predicates and then refines them with coordinate descent. During initialization, we first use `OptimizeWandPhi` to find a continuous representation $\tilde{\phi}_k$ for all predicate variables and then discretize each of them with `Discretize`. During refinement, we optimize individual predicates and discretize them one at a time while fixing all others. Our full algorithm is in Algorithm 1.

---

**Algorithm 1** Our algorithm based on `OptimizeWandPhi` and `Discretize`.

---
1: **Arguments**: $T, M$ // # iterations, # predicate candidates
2: **Output**: var_$\phi$, $\hat{w}$ // the current list of natural language predicates, other weights
3: // var_$\phi$ maintains a list of natural language predicates to be optimized and returned.
4: $\hat{\mathcal{L}}, \hat{w}, \tilde{\phi} = \texttt{OptimizeWandPhi}(\text{Empty\_Set})$ // relax during initialization
5: **for** $k = 0$ to $K - 1$ **do**
6:     $\hat{\phi}_k \leftarrow \texttt{Discretize}(\tilde{\phi}_k, c_k, M)[0]$ // discretize during initialization
7: **for** $t = 0$ to $T - 1$ **do**
8:     **for** $k = 0$ to $K - 1$ **do** // enumerate all $K$ parameters for $T$ rounds of coordinate descent
9:         $\tilde{\phi}_k \leftarrow \texttt{OptimizeWandPhi}(\text{var\_}\phi_{-k})$ // relax during refinement
10:         $H \leftarrow \texttt{Discretize}(\tilde{\phi}_k, c_k, M)$ // discretize during refinement
11:         **for** $h \in H$ **do**
12:             **if** $\texttt{OptimizeWandPhi}([\text{var\_}\phi_{-k}; h]).\hat{\mathcal{L}} < \texttt{OptimizeWandPhi}(\text{var\_}\phi).\hat{\mathcal{L}}$ **then**
13:                 var_$\phi[k] \leftarrow h$ // Update the $k^{\text{th}}$ parameter to $h$ if swapping it decreases the loss.
14: $\hat{w} \leftarrow \texttt{OptimizeWandPhi}(\phi).\hat{w}$ // fixing all $\phi$ and optimize $w$
15: **return** var_$\phi$, $\hat{w}$

---

## 5 EXPERIMENTS

We evaluate our algorithm on three datasets for each of the four statistical models from Section 3.2. In each dataset, samples are paired with reference descriptions. We used these samples and descriptions to generate modelling problems where recovering the reference descriptions leads to optimal performance. This allows us to evaluate our algorithm by computing the similarity between the reference and the predicted predicates. Through extensive ablations, we found that continuous relaxation, refinement, and steering with constraints are crucial to the performance. Finally, Section 5.4 presents proof-of-concept applications of controllable and explainable image clustering and explaining major topical variations across months, showing the potential versality of our NLCP framework.

### 5.1 DATASETS

We ran our experiments on three real datasets: New York Times (NYT) (Sandhaus, 2008), AG News, and DBPedia (Zhang et al., 2015); due to space limit, our main paper focuses on the NYT dataset and presents other results in Table 2. We chose these datasets because they all have annotated reference label descriptions $\vec{\phi}_*$ for each sample: In NYT, there are in total 9 topic labels, e.g. "*has a topic of estate*" and 10 location labels, e.g. "*has a location of Canada*", and each article is manually annotated with one topic and location label. We use NYT articles to generate modelling problems where recovering the reference $\vec{\phi}_*$ and their correct denotations lead to optimal performance.

**Contrastive Feature Learning.** We created 50 groups of articles, where each group has on average 30 articles with the same topic and location label, e.g. one group only contains articles about estate news in the U.S. Since there are in total 9 topics and 10 locations, we set the model parameter $\vec{p}$ to have $9 + 10$ dimensions; 9 have the constraint of "*being about the topic of an article*" ($c_{\text{topic}}$) and 10 have the constraint of "*being about the location of an article*" ($c_{\text{location}}$). The optimal parameters would model each group as $p(x|[\infty, \infty], [\text{"has the topic of \dots ", "has the location of \dots "}])$.

**Clustering.** We sample 1,500 articles and cluster either based on $c_{\text{topic}}$ or $c_{\text{topic}}$, and we set $K = 10$ under $c_{\text{location}}$ and $K = 9$ under $c_{\text{topic}}$.

Appendix A presents details about our topic modelling and regression problems.

### 5.2 METRICS.

We evaluate our algorithm by comparing the estimated parameters $\hat{\phi}$ and the reference $\vec{\phi}_*$. We consider two types of similarity: surface similarity and denotation similarity.

**Denotation Similarity.** For each pair of predicate $(\phi_1, \phi_2)$, we can quantify their denotation similarity by computing the F1 score of using $\phi_1(x)$ to predict $\phi_2(x)$ on the training set $B$. To compute the similarity between $\hat{\phi}$ and $\phi_*$, for each dimension $k$ in the reference, we find the most similar predicate in the estimated parameter and compute its F1 score; we then report the average across all $k$. Similar evaluation protocol is standard in the clustering literature when the ground truth is available (Lange et al., 2004; Wang et al., 2023).

**Surface Similarity.** For each reference predicate, we 1) find the predicate in the estimated parameter with the highest denotation similarity, 2) ask GPT-4 to rate their similarity, and 3) report the average of GPT-4 ratings. The similarity score is 100 if they have exactly the same meaning, 50 if they are related, and 0 if unrelated. Figure 3 "similarity judgement prompt" shows our prompt for GPT-4.

### 5.3 RESULTS

We investigate the following questions through ablation studies:

1. Does "continuously relax + discretize" outputperform searching for random predicates?
2. What types of embedding $e_x$ leads to the best performance?
3. Does the model successfully make use of the constraint $\vec{c}$?
4. Does iterative refinement improve the performance?

To execute our experiments, we use `google/flan-t5-xl` (Chung et al., 2022) to compute $\vec{\phi}(x)$, `gpt-3.5-turbo` (Ouyang et al., 2022) to extract constrain-relevant information $\Omega_c(x)$ and pro-

| Method | cluster_loc | cluster_topic | topic_loc | topic_topic | CFL | regr |
|---|---|---|---|---|---|---|
| RANDOM | 17/ 0 | 18/ 0 | 17/ 0 | 18/ 0 | 18/ 0 | 18/ 0 |
| NOCONSTR | 45/42 | 33/ 6 | 24/ 3 | 30/22 | 28/20 | 34/24 |
| NOREFINE | 33/39 | 44/34 | 45/16 | 53/18 | 28/36 | 29/29 |
| RANDOM_H | 34/45 | 13/15 | 30/16 | 46/12 | 11/11 | 13/12 |
| ONEHOTEMB | 42/39 | 47/39 | 52/28 | 69/32 | 28/20 | 34/26 |
| DIRECTEMB | 63/67 | **63/53** | **72/40** | 76/34 | **52**/52 | 40/40 |
| PROJEMB | **74/84** | 62/**53** | **72**/33 | **78**/36 | 50/**53** | **47/52** |

Table 1: Each value on the left/right of "/" means denotation/surface similarity. Each column is named with "model name _ constraint-type" (if any). RANDOM refers to a random baseline that computes the F1 similarity after shuffling the labels for each datapoint.

pose natural language predicates in `Discretize`, `hku-nlp/instructor-large` (Su et al., 2022) to create the sample embeddings $e_{\Omega_c(x)}$. We set $M$ and $T$, the number of predicates per `Discretize` call and the number of refinement iterations, to be 3. We report all experiment results in Table 1, where the last row PROJEMBED represents our full method. We now answer all four questions listed above by comparing PROJEMBEDto other variants.

**Our method outperforms randomly proposed predicates.** We compare to a baseline, RANDOM_H, that uses a `Discretize` function that returns $H$ by prompting the language model to generate predicates based on random samples from $B$. Comparing RANDOM_H and PROJEMBED in Table 1, we find that the later is better in all cases, indicating that our procedure is better than proposing predicates based on random samples randomly.

**Embedding $x$ with pretrained models improves the performance.** We compare three types of embddings mentioned in `OptimizeWandPhi` (Section 4): 1) ONEHOTEMB embeds $x$ with one-hot encoding, 2) DIRECTEMB embeds $x$ with a pretrained model, and 3) PROJEMBED extracts constraint-related information before embedding with a model. Table 1 shows that using pretrained embeddings is always better; projecting $x$ to extract constraint-relevant information sometimes leads to the best performance, while always performing at least comparably to all other approaches.

**The constraints steered the model learning process.** We compare to a baseline, NOCONSTR, where the `Discretize` oracle ignores the constraint $c_k$ and returns arbitrary predicates. Table 1 shows a performance decrease, so we conclude that steering with constraints is successful.

**Refinement improves the performance.** We consider a baseline, NOREFINE, which does not refine the predicates initialized at $t = 0$. Table 1 shows that refined predicates are always better.

Appendix Table 2 shows that the above claims can be reproduced on two other datasets, AG News and DBpedia (Zhang et al., 2015). Finally, our model-agnostic optimization framework still has a large room of improvement and lags behind methods developed for specific models. Appendix Table 3 reports a direct comparison with a prior work that focuses on clustering with NLCP (Wang et al., 2023), and we found that our method still significantly lags behind on DBPedia topic clustering.

## 5.4  PROOF-OF-CONCEPT NLCP APPLICATION.

We present two proof-of-concept applications: 1) controllable and explainable image clustering and 2) explaining topical variation across months. In contrast to the experiments above, these results are cherry-picked to demonstrate the potential of our NLCP framework and not rigorously evaluated.

**Image Clustering.** Unsupervised image clustering is widely used to discover new objects (Sivic et al., 2005) or explore datasets (Caron et al., 2018), but these methods might not be easily steerable or explainable. In this section, we create image clusters with associated explanations and steer them with natural language constraints. As shown in Figure 2 top-left, we perform clustering on a synthetic image dataset sampled from DomainNet (Peng et al., 2019), where there are three styles (painting, real, and sketch) and three objects (bird, flower, and bicycle), with 16 images for each of the $3 \times 3$ combinations. The clustering constraint is either to "*cluster by style*" or "*cluster by object*". To optimize the loss of the clustering model, we use ViLT (Kim et al., 2021) to approximate the denotation of a predicate on an image and propose predicates by prompting `gpt-3.5-turbo` based on image captions generated by BLIP-2 (Li et al., 2023).

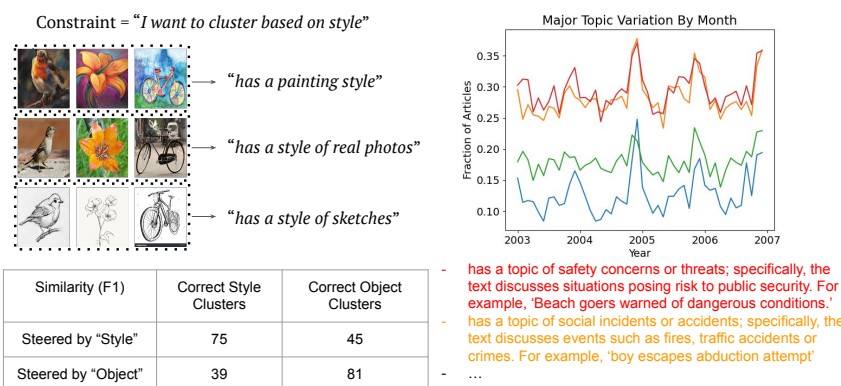

Figure 2: Two proof of concept application of NLCP. **Left**: we generate explainable image clusters based on natural language constraints. We found that when steered with the style/object constraint (row), the generated clusters are more similar to the reference style/object clusters (column).**Right**: we explain major variations across different months with detailed natural language predicates and plot how often each predicate is true across different months.

Figure 2 bottom-left displays results when steering with the style or object constraint, respectively. The predicted clusters are more similar to the reference style/object clusters, indicating that steering with natural language constraint is useful. This demonstrates that NLCP can be applied to the image modality, unlocking a wide range of image-based data analysis opportunities, e.g. explaining what types of images different groups of users might share on social media.

**Explaining Major News Topical Variations Across Months.** Many applications could benefit from explaining the differences between a large set of corpora in natural language, e.g. understanding topical differences across many months of news titles, user preferences in their reviews across thousands of different zip-code, etc. We provide one such example by explaining the major topical variation across 48 months of news titles from Australia Broadcasting Corporation (ABC)[2], using the topic model ($K = 8$) under NLCP, and plotting how often various learned predicates are true for each month (Figure 2). This demonstrates that NLCP could unlock new application opportunities.

## 6    CONCLUSION AND FUTURE WORK

We proposed a framework, NLCP, to improve the explainability and steerability of statistical models (Section 3). To optimize the likelihood in NLCP, we introduced continuous relaxations and iterative refinement (Section 4), and validated these ideas for several datasets and NLCP models (Section 5).

However, there is still significant room for improving NLCP models and methods. First, although our paper proposes and validates a few effective model-agnostic optimization techniques, we still underperform model-specific optimizers (Table 3). A competitive model-agnostic NLCP algorithm would allow practitioners to explore new models without worrying about optimization details, similar to how auto-diff libraries allow deep learning practitioners to explore new architectures.

Second, our work focused on the text domain, while language can also describe other modalities, such as vision or sound; to extend NLCP to those modalities, we need stronger foundation models to propose predicates and validate them on the modality of interest.

Finally, while we explored several statistical models, there is a large space left to explore. For instance, we only used natural language parameters for one round of feature extractions, while one can model hierarchical structures across natural language parameters (e.g. a taxonomy where each node is represented by a predicate). By building NLCP models with richer structures, it is possible to build more expressive models while still maintaining the intuitive natural language interface between ML systems and humans, and thus minimize the performance gap between uninterpretable systems vs. explainable and steerable systems.

---

[2]https://www.kaggle.com/datasets/therohk/million-headlines.

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

# A  DATASETS FOR OTHER MODELLING TASKS

**Topic Model.** We create a separate test bed for modelling texts as a mixture of texts with the same location ($c_{\text{location}}$) or a mixture of topics ($c_{\text{topic}}$). For each constraint, we generate 30 groups of articles; for each group $n$ we sample a mixture probability distribution $w_n$ from a dirichlet distribution with parameter = 1 over all labels under that constraint; then we sample 75 articles by first sample a label $l$ from $w_n$ and then uniformly sample an article conditioned on $l$. We set $K = 10$ under $c_{\text{location}}$ and $K = 9$ under $c_{\text{topic}}$. The optimal parameters would disentangle the mixtures and uncover the latent "topics" represented by predicates.

**Regression.** We sample 1,500 articles $x_i$ and create a target vector $y_i$ for each of them. To do so, we first associate each of the 19 location or topic labels with a 32 dimensional vector sampled from $\mathcal{N}(0, I)$; then for each $x_i$, we set $y_i$ to be sum of its location vector and the topic vector. We set $K = 19$, where 9 constraints are $c_{\text{topic}}$ and 10 are $c_{\text{location}}$.

# B  MORE EMPIRICAL RESULTS

In Table 2, we performed the same set of clustering and topic modelling experiments from Table 1 on AG News and DBpedia Zhang et al. (2015). We did not perform the low-rank factorization and the regression experiments since they require more than one types of labels for each sample to be meaningful, while AG News and DBpedia only contains topic labels.

|            | cluster_AG | cluster_DBpedia | topic_AG | topic_DBpedia |
|------------|------------|-----------------|----------|---------------|
| RANDOM     | 29/ 0      | 14/ 0           | 29/ 0    | 14/ 0         |
| NOREFINE   | 69/72      | 26/20           | 67/75    | 39/34         |
| RANDOM_H   | 32/37      | 14/11           | 47/50    | 26/19         |
| ONEHOTEMB  | 58/55      | 31/19           | 60/65    | 24/20         |
| DIRECTEMB  | 80/72      | 36/33           | **81/89**| 38/37         |
| PROJEMB    | **82/80**  | **40/30**       | 80/87    | **42/36**     |

Table 2: Models' performance on other datasets AG News and DBPedia (Zhang et al., 2015). We observe a similar set of conclusions as in Section 5.3.

Table 3 reports a head-to-head comparison with prior state-of-the-art on explainable clustering (Wang et al., 2023), which solves the same clustering task as ours but designs a task-specific optimization algorithm based on integer linear programming. Our algorithm is comparable to the state-of-the-art method on some datasets (AG News, NYT_topic) but still lags behind on others, indicating a large room for future improvement for model-aganistic optimization methods. On the other hand, it significantly outperforms LDA, and respects the constraint more than pure neural embedding-based method (Instructor) when clustering based on locations.

| Method     | NYT_top (9) | NYT_loc (10) | AG News (4) | DBPedia (14) |
|------------|-------------|--------------|-------------|--------------|
| LDA        | 51/ 0       | 40/ 0        | 53/ 0       | 51/ 0        |
| INSTRUCTOR | 69/ 0       | 56/ 0        | 84/ 0       | 82/ 0        |
| PAS        | 70/67       | 76/100       | 87/88       | 71/54        |
| OURS       | 61/53       | 66/ 84       | 82/80       | 51/30        |

Table 3: Each column corresponds to a clustering dataset, with "{dataset_name} {constraint_name} ($K$)", and the default is to cluster based on topic if the constraint is not provided. The methods above the line (INSTRUCTOR (Su et al., 2022) and LDA (Blei et al., 2003)) do not provide natural language explanations; PSA (Wang et al., 2023) solves the exact same clustering task as our method but involves an optimization specifically designed for clustering. Our algorithm still lags behind PAS, indicating that the optimization procedure still has a significant room for improvement; on the other hand, our method significantly outperforms LDA in most cases and can respond to different constraints (e.g. cluster by locations/topics).

## C  PROMPTS

We show the prompts used in our experiments in Figure 3. The language model predicted outputs are highlighted in light blue.

### Discretizer prompt

**Constraint**

Here is a corpus of text samples each associated with a score. The text samples are sorted from the lowest to the highest score.

I want to understand what topic of text achieves a higher score. Your response should start with "has a location of" ….'

**Samples from B**

1. "athlete demonstrated remarkable prowess." (score: -0.2)
2. "see the player last night?" (score: -0.3)
3. "the musician resonated with profound emotional undertones" (score: 0.3)
4. "Wonderful paining by " (score: 0.4)

We want to understand what kind of text samples achieve a higher score, so please suggest descriptions about the text samples that are more likely to achieve higher scores.
 - "uses double negation"
 - "has a conservative stance; specifically"

Please generate the response based on the given datapoints as much as possible. We want the descriptions to be relatively objective and can be validated easily. For example…
Your responses are:

**Language Model Outputs**

- "has a casual style"

### Similarity judgement prompt

Is text_a and text_b similar in meaning? respond with yes, related, or no.

Target:
text_a: has a casual style
text_b: is in an informal style
output: **yes**

### Denotation prompt

Check whether the TEXT satisfies a PROPERTY. Respond with Yes or No. When uncertain, output No.

Now complete the following example -
input: PROPERTY: has a casual style
TEXT: "see the player  last night?"

output: **yes**

### Project prompt

In this task the user wants to extract some key information from a text. Come up with a few key phrases (in English) based on the text based on the goal.

constraint: We want cluster the text based on their topics.

text: the musician resonated with profound emotional undertones

keyinfo (in English): **art, music, performance**

Figure 3: We show the prompts used in our experiments. The language model predicted outputs are highlighted in light blue.

## D  DEGENERATED OPTIMAL SOLUTION OF MIXTURE MODEL

We consider a Gaussian Mixture Model for the clustering example, where we marginalize over all clusters to obtain the probability for each datapoint $x$ and compute the sum of their log probability. When all clusters $p_k := B(x)$, a uniform distribution over all training samples, we achieve the lowest negative log loss.

$$\mathcal{L}(\vec{p}) = -\sum_{x \in X} \log(\sum_{k=1}^{K} p_k(x)); \quad p_k(x) := p(x|[\tau], [\vec{\phi}_k]), \tau \to \infty \tag{16}$$

Why did this not happen for Gaussian mixture models? Let's consider a data distribution $p$ generated by a mixture of two different Gaussian variables $\mathcal{N}(\mu_1, \sigma_1)$ and $\mathcal{N}(\mu_1, \sigma_2)$: to fit $p$ with two gaussian variables, the optimal solution is to indeed recover the two underlying gaussian variables in the generative process, and fitting the mixture with one gaussian would lead to suboptimal solution.

However, since natural language predicate can easily express a union relation of two predicates, it is often also optimal to fit the mixture of two predicate-parameterized distribution with one distribution. For example, in the domain of articles that only have 2 different topics "*sports*" and "*arts*", the predicate of "*True*" or "*articles*" express the union over the two clusters and hence can perfectly model the mixture distribution of two sub-clusters.

