# OpenReview forum: "Explainable, Steerable Models with Natural Language Parameters and Constraints"
_ICLR.cc/2024/Conference — Submitted to ICLR 2024_

### Official Review · Reviewer_53NP · 2023-10-27

**Soundness:** 2 fair
**Presentation:** 1 poor
**Contribution:** 2 fair
**Rating:** 3
**Confidence:** 4

**Summary:**

This paper considers the problem of controlling classical statistical models with natural language constraints. The paper defines natural language predicates which are learned from a simple linear model combined with prompting an LLM to describe the individual components of the linear model by observing the scores each component gives to each example. The predicates produced can be steered by the user providing a meta-constraint, a predicate on that should be satisfied by all of the predicates. Various ablations are evaluated empirically against one another using a LLM evaluation scheme contrastive feature learning and clustering datasets.

**Strengths:**

* This paper presents an interesting idea of using an LLM to generate natural language predicates to explain predicates which guide model predictions

**Weaknesses:**

* The writing needs to be improve. The paper is written in a way that seems to hide the underlying mechanism used to generate the predicates.
* The experimental results are severely lacking. There is a lack of volume, baselines, credible evaluation metrics.
* The correctness of the proposed approach is highly dependent on the quality of the LLM used.

**Questions:**

* Is the point of this paper to use an LLM to generate explainable predicates? Is there another way in which optimization could be done to generate such predicates? What if the modeling decisions do not yield explainable predicates?
* The paper seems to be trying to solve to many types of problems and being too general. This seems to be creating quality issues.
* How can you evaluate whether a predicate satisfies a constraint? There cannot be any guarantees about whether predicates can satisfy the constraints correct?
* How could this approach be used in an iterative human-in-the-loop fashion? Does the method allow for utilizing the previous predicates when incorporating new constraints?

---

### Official Review · Reviewer_th8x · 2023-11-01

**Soundness:** 2 fair
**Presentation:** 3 good
**Contribution:** 3 good
**Rating:** 5
**Confidence:** 3

**Summary:**

The authors investigate the important problem of improving explainability and steerability in statistical models. The authors propose an innovative framework that uses natural language as hyper-parameters, such that the rich semantics in texts can be effectively leveraged. The algorithm is comprehensively evaluated across three real corpora and four statistical models. Furthermore, the authors provide demonstrations of proof-of-concept applications in controllably generating explainable image clusters and describing major topic variations.

**Strengths:**

1. The paper is well-written and easy to follow.

2. The authors propose a novel framework that innovatively leverages the denotational semantics of natural language.

3. The authors provide demonstrations of proof-of-concept applications in controllably generating explainable image clusters and describing major topic variations.

**Weaknesses:**

1. The authors only conduct experiments on three datasets, which can be insufficient for a fair evaluation of their method.

2. In the experiments, the authors seem not to utilize state-of-the-art baselines. Specifically, the baselines utilized in the experiments are not clearly demonstrated and thus result in difficulties in fairly assessing the effectiveness of the method. The authors do not provide details of these baselines, either.

3. The authors did not conduct experiments to further evaluate the effects of different parameters introduced in the algorithm. As a result, the experimental part is not well structured.

**Questions:**

Have the authors explored more challenging tasks as applications of the proposed work?

---

### Official Review · Reviewer_1mws · 2023-11-01

**Soundness:** 2 fair
**Presentation:** 2 fair
**Contribution:** 2 fair
**Rating:** 3
**Confidence:** 3

**Summary:**

Working towards the end goal of improving the explainability and steerability of machine learning models, the authors propose representing parameters as natural language predicates. In this framework, a natural language predicate such as "has casual style" could, for instance, take place of a Gaussian representing a cluster. To learn in this framework, the authors propose an algorithm which iteratively alternates between continuous relaxations of string parameters and their discrete counterparts obtained through querying a language model for the most likely explanation.

**Strengths:**

- I find the idea of using natural language sentences as binary feature extractors quite novel

**Weaknesses:**

- All of the tasks considered in the experimental section seem very toyish. It is, therefore, not very clear if the proposed method actually fits in the current state of machine learning.

- Relatedly, the approach seems to require a complete overhaul of current machine learning pipelines, which is not catastrophic, but makes me wonder how widely adopted it might become compared to an add-on method.

- The authors seem to rely extensively on prompting large language models for various steps throughout their approach, which first of all raises the question of how computationally feasible the proposed approach is, and secondly makes me doubt the extent of the paper's contribution due to how adhoc and unprincipled it seems.

- While the writing is not very hard to follow, I believe that a running example would greatly help the exposition of the paper.

**Questions:**

- My first question is a high-level one: to obtain the benefits of natural language predicates, why not simply assign these natural language predicates, e.g. to clusters, in a post-hoc fashion?

- Am I correct in my understanding that equation (5) assigns a zero probability to all sentences that do not have a sports topic and non-zero otherwise? Am I also correct in my understanding that this necessitates combing through entire dataset and prompting the language model whether each of the sentences therein satisfies the natural language predicate, and repeating that for every predicate $\phi_i$. That seems to already be very computationally expensive.

- I am not sure I understand the subsection "Modeling Uninformative Nuisances with a Base Distribution". How is the base distribution $B$ typically defined? Is it some pretrained model? Could you please expand on what you mean by "the background distribution is this important, but only for the purposes of selecting good $w$ and $\phi$" as well as your use of a "semiparameteric approach"

- In Section 4, how do you get the continuous unit vector $\tilde{\phi}\_k$ ? Is it simply the standard unit vector?

- I would've expected to see some form of gradient estimator here to back-propagate through the non-differentiable discretization. Could you please explain why that's not needed here?

---

### Official Review · Reviewer_xQLL · 2023-11-04

**Soundness:** 3 good
**Presentation:** 2 fair
**Contribution:** 2 fair
**Rating:** 5
**Confidence:** 4

**Summary:**

This paper suggests a framework for explaining ML models based on different aspects (e.g. topic, writing style (formal or informal)) using natural language. The framework is named natural language constraints and parameters (NLCP) and allows learning natural language parameters which are easy to learn and control.
The framework proposes an optimization algorithm for choosing the optimal explainable predicates \phi and their weights w. Since the explainable predicates are discrete the paper proposes a continuous relaxation through an activation mapping function.

The authors showcase the effectiveness of their approach on 3 different datasets and 4 statistical models (Clustering, Contrastive Learning, Regression and Topic Modelling). As evaluation metrics, the authors use denotation and surface similarity scores between predicted and reference predicates. They use GPT-4 to judge the similarity and compare with their prediction.

**Strengths:**

The paper proposes an interesting idea of providing common explanations using natural language based on different criteria. For example, in the case of image clustering this approach allows to explain cluster nodes based on different criteria such as the style of the images or the type of the objects in the images.

**Weaknesses:**

1) Learned parameter explanations on Figure 1 don’t seem very clear. It would be good to explain them. The visualizations for Topic models and Contrastive Feature Learning look the same in Figure 1.
2) The contributions of the paper are not clear. It would be good to list the contributions clearly in the introduction section.
3) Overall, mathematical notation looks confusing. [[c]] (\phi) and [[\phi]] (x) are difficult to follow. It looks like \phi = “ has a casual style” and an example of  c is “the predicate should be style- related”. The roles of \phi and c constant are difficult to distinguish and follow in the beginning of the paper. Later, in the experimentation section they become more clear.
4) OptimizeWandPhi and Discrete do not sound theoretically grounded. It would be good to strengthen the theoretical justification of the optimization method.
5) A visual representation of the algorithm could be helpful. Currently, the algorithm listing is not very clear.

**Questions:**

1) How can we ensure that our evaluation is reliable and representative ? How can we ensure that GPT-4 provides a reliable rating for the similarity ?
2) How do we come up with constraints such as 	“I want to cluster based on style” ? How can we ensure that we have a representative set of constraints ?

---

### Meta-Review · Area_Chair_FntC · 2023-12-08

**Metareview:**

**Paper Summary:**

The paper presents an approach to enhance the explainability and steerability of machine learning models by integrating natural language predicates. The authors introduce an algorithm to select optimal explainable predicates and their corresponding weights, incorporating a continuous relaxation and discretization mechanism. The effectiveness of this approach is evaluated using three different datasets and four statistical models, including clustering, contrastive learning, regression, and topic modeling.

**Strengths:**

1. Novelty: The idea of using natural language sentences as binary feature extractors is novel (xQLL, 1mws, th8x, 53NP).

**Weaknesses:**

1. Lack of Comprehensive Experiments: The experiments conducted are "toyish" (1mws), and lack comparison with state-of-the-art baselines (th8x, 53NP).
2. Reliance on LLM performance: The correctness of the approach relies on the quality of the LLM (53NP, 1mws)
3. Lack of Clarity: While the paper is overall easy to follow, there are some parts that are confusing (xQLL, 1mws, 53NP)

**Decision:**

While the concept and approach are innovative and show potential, given the concerns raised by the reviewers, I do not recommend its acceptance.

**Justification For Why Not Higher Score:**

I'm not recommending the acceptance of this paper due to the above concerns raised by reviewers.

**Justification For Why Not Lower Score:**

N/A

---

### Decision · Program_Chairs · 2024-01-16

Reject